# PBES: PCA Based Exemplar Sampling Algorithm for Continual Learning

## Abstract

Traditional machine learning is both data and computation intensive. The most powerful models require huge quantities of data to train and the training is highly time consuming. In the streaming or incremental model of machine learning, the data is received and processed in a streaming manner, i.e., the entire data stream is not stored, and the models are updated incrementally. While this is closer to the learning process of humans, a common problem associated with this is "catastrophic forgetting" (CF), i.e., because the entire data is not stored, but just a sketch of it, as more and more data arrives, the older data has invariably a smaller representation in the stored sketch, and this causes models to perform badly on tasks that are closer to older data. One of the approaches to solve this problem stores an "exemplar set" of data items from the stream – but this raises the central question: how to choose which items to store? Current approaches to solve this are based on herding, which is a way to select a random looking sample by a deterministic algorithm. We propose a novel selection approach based on Principal Component analysis and median sampling. This approach avoids the pitfalls due to outliers and is both simple to implement and use across various incremental machine learning models. It also has independent usage as a sampling algorithm. We achieve better performance compared to state-of-the-art methods.

## 1 Introduction

In traditional machine learning, one usually trains a model from a given data set that serves for training and testing purposes. The model can then be applied to various queries. On the other hand, humans learn and build "mental models" incrementally over time from a series of experiences and information sources. The corresponding machine learning scenario is known as *continual learning* (CL). In this scenario, a machine learning model continually keeps learning and adapting to new data; in effect, the data is viewed as a *stream* rather than a batch. Both humans and continual learning systems suffer from the problem of forgetting. Humans can forget information they learned in the past, whereas a machine learning model's adaptation to new incoming data suffers from so-called *catastrophic forgetting* (CF) due to inaccessibility of the data of earlier tasks in CL setups. In machine learning, a trained model depends heavily on the class distribution seen in the data. The reason for CF is that the class distributions of incoming data changes over time. Newer classes are introduced with time and this invariably changes the class distribution. Conceptually the stream of data can be thought of as a sequence of *tasks* where each task usually consists of data about a few new classes. Thus, different tasks have different class distributions and the model needs to update itself in order to recognize the new classes. It is impractical to retrain a model over all the past data, and thus one "remembers" either just a sample of the older data or else one tries to make the model itself more complex so that it can hope to just incorporate the newer data into the additional model complexity. In either case, there is some forgetting involved as the entire old data cannot be stored or remembered in the model. The challenges posed by CF cause difficulties for AI models to adapt to practical systems across a multitude of fields (Prabhu et al., 2020; sem; Li et al., 2020). In particular, different streams of data have different inter-task distributional gaps and thus pose challenges of different natures. While there have been continual learning studies on various domains such as image classification on commonly found day-to-day objects (CIFAR10, CIFAR100, ImageNet datasets), food images (Food1K dataset), but datasets with high diversity (large inter-class gaps) have not been studied previously. Such data usually has outliers, and it

becomes crucial to remember data that is not an outlier. An example domain that has this large data variance is the sports domain. In the sports domain, there is high diversity and a large inter-class gap, and thus existing approaches do not perform well while classifying images of various sports. Another difficulty associated with learning systems is handling class imbalance. Labeled data for classes may be highly imbalanced and this affects the model trained so that its performance can be bad. This problem is perhaps exacerbated in a continual learning scenario where only a sample of the older data can be remembered. In case the data is highly imbalanced, it is possible that the rare classes get no representation in the sample at all, while the dense classes occupy all of the samples. This can potentially lead to overfitting or underfitting of the trained model. The goal of this paper is to address these problems. First, we want to design a more robust sampling scheme so that the data remembered is less prone to outliers. Perhaps an example real-life scenario will serve to convey the problem. Imagine attending a party at a large company you just started working for. At the party, you meet many people, but you want to retain the names of the "key" people. In particular, if you cannot remember everyone's name, perhaps it is good to remember those of the people in your group or closely related ones as well as the top company officials, but compromise on those of their friends or family members they may have invited. Second, our learning approach should be able to handle class imbalance gracefully. There have been intriguing findings in recent rehearsal-based CL papers – approaches that maintain a fraction of previously seen data when training new incoming classes (Prabhu et al., 2020) in class-incremental scenarios, hence, mitigating CF (Goodfellow et al., 2013). However, as mentioned, in rehearsal-based class-incremental scenarios an important question arises: how should the representative memory be managed optimally? Due to the limited number of stored data points compared to the number of incoming data points, during training the stored data points could either suffer from over-fitting or be disregarded because of the large quantity of incoming data points. A naive approach would be to progressively raise the storage size as new tasks are coming; however, this technique neglects an important representative memory constraint, i.e., to store a fixed number of data points. Hence, an approach is required that can preserve enough information about the previous class while using a modest number of data points.

The literature (Castro et al., 2018; Rebuffi et al., 2017; Wu et al., 2019; Zhao et al., 2020; He et al., 2020; Hou et al., 2019) mainly uses a herding algorithm (Welling, 2009) for choosing data points, also known as *exemplars*, that is based only on the class mean. As per Javed & Shafait (2018), herding algorithm is no better than a random selection of data points. Many researchers have proposed other effective algorithms to select data points or exemplars in rehearsal based method to mitigate the CF (Kim et al., 2020; Aljundi et al., 2019a; Chen & Lin, 2021; Wiewel & Yang, 2021). As mentioned before however, none of the current approaches performed well in our experiments when the data variance is very large, such as in the sports domain. We propose a novel sampling algorithm that performs better than state-of-the-art methods in CL. Our proposed continual learning system is effective for both class-balanced and class imbalanced datasets. The proposed system is effective even when the dataset is sparse and intra-class variation is high. To test the performance of our system in a class imbalanced scenario, we use it for the image classification problem in the sports domain. For our experiments we have used Sports73, Sports100 and Tiny ImageNet datasets. See Figure 1 for a sample of images from Sports100 dataset. Our main contributions are as follows: (1) A novel sampling algorithm PBES to select exemplars is proposed that is robust even when outliers are present. (2) We show how to mitigate class imbalance issue in continual learning settings by using KeepAugment (Gong et al., 2021) – a data augmentation approach. (3) We demonstrate the efficiency of our proposed method using two class-imbalanced Sports73 and Sports100 image datasets and one balanced Tiny ImageNet dataset. We demonstrate that our overall continual learning system outperforms existing state-of-the-art approaches in all cases.

## 2 RELATED WORK IN CONTEXT

In this section, we mention some related work and how the work in this paper fits in the context. Intuitively, CL can be viewed as learning where the data is presented as a stream during the learning phase. Even so, it can be divided into tasks where each task is a group of related data. For concreteness, let us consider a classification scenario. In this case, a task is a data for a sequence of classes. Data for a class in a task is not repeated in future tasks. There are two main situations under which continual learning works (van de Ven et al., 2021) – (i) task-incremental (TIL): here a *task ID* is present along with class data. This comes of use later in testing, where during a classification query, the task ID of the (unknown) class is available to us, and, (ii) class-incremental (CIL): no such task

ID is present. CIL is thus more challenging than TIL, and this paper is about CIL. Even though there is no task ID, we may conceptually divide the data stream into into chunks where each chunk has data for a fixed number of classes. Each such chunk can be termed as a task. Incremental learning methods can be divided into three approaches (Mai et al., 2022; He & Zhu, 2021; Luo et al., 2020): 1) rehearsal-based approaches: where representative memory can be used to retain some exemplars of previous tasks, which can subsequently be replayed in the current task (Wan et al., 2022; Yan et al., 2022; Chaudhry et al., 2018; Castro et al., 2018; Rebuffi et al., 2017), 2) regularization-based approaches: where no exemplars from previous tasks are kept, but the information from the model parameters is used to train model on new incoming tasks (Kirkpatrick et al., 2017; Zenke et al., 2017; Liu et al., 2018; Li & Hoiem, 2017; Lee et al., 2017; Mallya et al., 2018), and, 3) architecture-based approach which involves continuously expanding the model to fit the new incoming tasks without affecting the prior network parameters. As per literature (Mai et al., 2022; He & Zhu, 2021; Luo et al., 2020), oftentimes, replay-based approaches outperform regularization-based approaches in CIL setups. Assuming representative memory exists, the aim of this paper is to improve representative memory management for storing exemplars. We also examine the significance of exemplars memory size using a novel algorithm for exemplars selection. We now look at two important issues that arise during CIL using a rehearsal-based approach.

**Class Imbalance in training data:** The techniques in rehearsal-based approaches have documented significant catastrophic forgetting as a result of an imbalance in the class's data points (Wu et al., 2019). Models are sensitive to the most prominent memory buffer classes. The gradient information is used to update the model's weight from the previous tasks in MER (Riemer et al., 2018), GSS (Aljundi et al., 2019b), and GEM (Lopez-Paz & Ranzato, 2017). In the recent past, MEGA (Guo et al., 2020) has proposed a loss-balancing strategy to alleviate forgetting by combining the loss of prior and present classes. Chaudhry et al. (2020) suggest using the most damaging examples from previous tasks as anchor points to solve the CF problem (the HAL approach), whereas Balaji et al. (2020) suggest retaining more details by saving intermediate activations in conjunction with the raw images (the CAL approach). Nonetheless, these methods disregard the significance of representative memory management and often employ ring-buffer (Chaudhry et al., 2020) or reservoir sampling (Riemer et al., 2018), or simple random sampling (Balaji et al., 2020; Guo et al., 2020).

**Representative Memory Management:** Here memory management refers to managing the *content* of the memory, i.e., which exemplars to store. Various techniques are found in the literature (Parisi et al., 2019) for this purpose. Intriguingly, considering their computational complexity, a number of suggestions (Rebuffi et al., 2017; Chaudhry et al., 2018; Castro et al., 2018) demonstrate only minor accuracy improvements above regular random sampling. Discriminative sampling (Liu et al., 2020), herding selection (Welling, 2009), and samples based on entropy (Chaudhry et al., 2018) are a few examples of these approaches. Herding selection selects samples proportionate to the number of data points in each class, depicting each data point's distance from its class mean. Discriminative sampling selects data points which establish decision boundaries between classes. In entropy-based sampling, data points are selected based on the randomness of their softmax distributions in the final layer. Each class distribution's mean and boundary may be accurately represented by Liu et al. (2020) suggested sampling approach. Using a cardinality-constrained bilevel optimization approach, a representative memory technique generation of a coreset is given by Borsos et al. (2020). A GAN-based memory is suggested by Cong et al. (2020) that can be perturbed for exemplars in incremental learning. These papers address the effectiveness of the data points kept in memory; nonetheless, training a sample GAN for exemplars' memory would be computationally intensive or complex (Borsos et al., 2020). Because of the instability of GAN-based sample generation, we must instead rely on effectively sampling exemplars.

## 3 NOTATION, FORMAL SETUP, AND BASIC ARCHITECTURE

During CIL training, a stream of data is presented to the learning system and this can be divided into $T$ tasks. Conceptually, each task is a pair $t = (\mathcal{C}, \mathcal{D})$ of a set of classes $\mathcal{C}$ and data about them $\mathcal{D}$, so overall the stream looks like, $(\mathcal{C}^1, \mathcal{D}^1), (\mathcal{C}^2, \mathcal{D}^2), \ldots, (\mathcal{C}^T, \mathcal{D}^T)$, where $t_i = (\mathcal{C}^i, \mathcal{D}^i)$ is the $i$th task. We let $\mathcal{C}^i = \{c_1^i, c_2^i, \ldots, c_{m_i}^i\}$ where the $c_j^i$ are the $m_i$ classes in the set $\mathcal{C}^i$. The dataset $\mathcal{D}^i$ is the collection of points, $\{(x_1^i, y_1^i), (x_2^i, y_2^i), \ldots, (x_{n_i}^i, y_{n_i}^i)\}$ where $x_j^i$ is a data point and $y_j^i \in \mathcal{C}^i$ is its

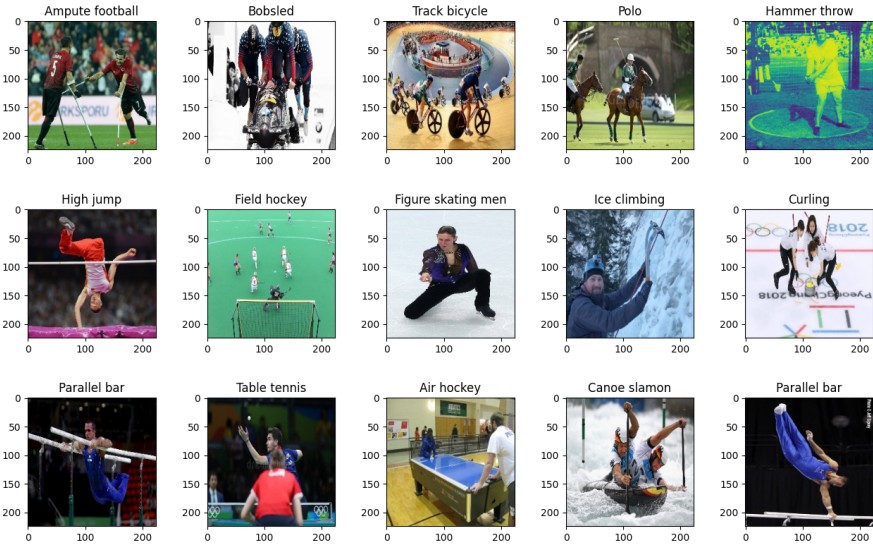

Figure 1: Random images from Sports100 dataset

label. Thus, the total number of data points is, $\sum_{i=1}^{T} n_i$ and the total number of classes is $\sum_{i=1}^{T} m_i$. Notice the classes are disjoint across tasks, so $\mathcal{C}^i \cap \mathcal{C}^j = \emptyset$ for $i \neq j$.

Here we will assume for CIL that the number of classes $m_i$ does not change across the tasks so $m_i$ is same for all $i$, however the total number of data points can vary from task to task, so $n_i$ may not equal $n_j$ for $i \neq j$. Moreover, the number of data points in $\mathcal{D}^i$ that have $y_j^i = c_k^i$, i.e., are of the $k$th class, may not be the same for all $k$ between 1 and $m_i$. This is termed class imbalance.

We now describe briefly the architecture of most offline learning systems, including for example iCaRL (Rebuffi et al., 2017). The main components are (1) *Representative memory* which is used to store and manage data points that best represent what the model has learned in previous tasks. Storing all prior data points may be impracticable due to storage issues. To alleviate CF in class-incremental settings, rehearsal-based methods retain a small amount of previously seen data termed *exemplars*. Two important aspects of representative memory are *selection and discarding techniques* which refers to which examples are stored as exemplars and which of them are discarded, and the *memory size* which refers to how many exemplars are stored; (2) *feature extractor* - this is usually a deep neural network that has been trained on two different loss functions, such as distillation loss and cross-entropy loss, and it is typically called a cross-distilled loss function. It extracts a fixed number of features from every data point; (3) *weighting policies* - in real-world CL systems, some tasks may have more importance than others, whereas, during training, distribution techniques fail to take into account the significance of each task. Thus, it is important to consider all tasks with some weighting policy; (4) *classifier* - this is an algorithm that classifies a test data point into one of the classes seen so far; (5) *data augmentation* - this refers to increasing the number of data points by modifying the original dataset to alleviate the problem that only a small amount of data is available in representative memory for previously seen classes. This increases inference accuracy.

## 4 OUR APPROACH

Our proposed method to mitigate CF includes a novel technique of selecting exemplars. This technique is coupled with an effective data augmentation technique to produce a balanced training dataset from an imbalanced dataset. The overall workflow is shown in Figure 2. Here, F.E. and D.A. stand for Feature Extractor and Data Augmentation, respectively. Rather than choosing exemplars according to the class mean, as used in herding (Welling, 2009), we develop our custom sampler

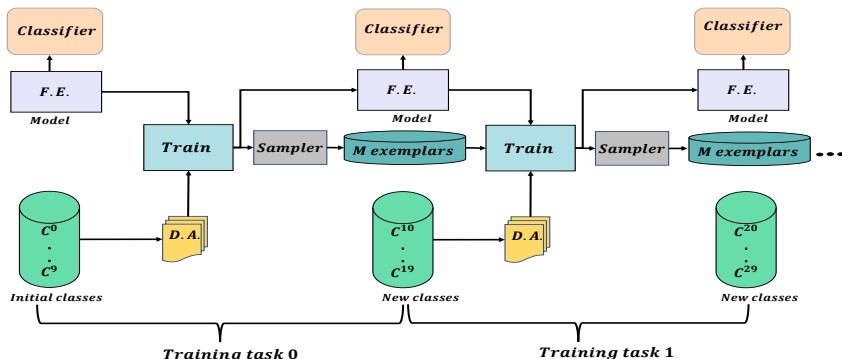

Figure 2: Architecture of Our Continual Learning Approach

which is based on Principal Component analysis and median sampling. The KeepAugment (Gong et al., 2021) algorithm is used to supplement each new incoming class, creating a balanced data stream. We apply the classification loss to this balanced stream and the distillation loss to the exemplars stored. In the next section, we will discuss each component's specifics.

### 4.1 SELECTION OF EXEMPLARS

In this section we present our algorithm PBES to select the exemplars. The pseudocode for the algorithm is shown in Algorithm 1. The input to the algorithm is a training set $X = \{x_1, x_2, \ldots, x_n\}$ of $n$ images of a particular class $c$ and a target integer $m \leq n$ of images to select as exemplars. Henceforth we simply call the images "points" as they are treated as points in a high-dimensional space. The algorithm starts by computing $p$ as $\lceil m/2 \rceil$ or $\lceil m/2 \rceil + 1$, the latter choice being taken only when $n$ is odd and $m$ is even. Then, it runs the Principal Component Analysis (PCA) algorithm on $X$ and computes the first $p$ principal directions. Next, it initializes a set $R$ to $X$. The set $R$ is the set of remaining points, none of which have been selected. The sequence of selected points is stored in $P$, of which the first $m$ will be returned at the end. It repeatedly does the following: projects and sorts the remaining set $R$ of points on the $i$th principal direction (for $i = 1, 2, \ldots$) and select either 1 (when $|R|$ is odd) or 2 (when $|R|$ is even) median points, and appends them to $P$, removing them from $R$. The following is elementary. For completeness a proof is included in Appendix A.

**Lemma 4.1** *The sampling algorithm appends exactly $m$ or $m + 1$ points to $P$ (before returning its first $m$ points), depending on whether $m, n$ have the same or different parity.*

**Intuition**  There are two key ideas involved in Algorithm 1. The first one is the selection of the median points along projection to some direction (this happens to be a principal direction). The reason for this is to make the selection of exemplars robust to outliers. A common approach is to select the exemplars by distance from the mean. However, since the mean can be affected greatly by the presence of outliers, this strategy does not perform well for datasets with outliers, for example Sports73 and Sports100 datasets, where variance is pretty high given the nature of sports data and outliers are to be expected. The second key idea is to project the data along the principal directions before doing this. To see why, imagine a dataset with intrinsically low dimensions in a high dimensional space. Suppose it lies on a low-dimensional affine subspace. Projecting the data to a direction orthogonal to this subspace will map all the points to the same point and the median will not have any meaning. Thus, only directions lying in the subspace really matter. For such directions, the "best" ones are those that expose the maximum variance in the data, as intuitively the outliers "cause" the high variance. The first principal direction is the best for this, and as the subsequent principal directions maximize the "residual" variance they are preferred. There is recent work in statistical sampling using similar ideas (Deng et al.).

Another idea, that we use as a control, is to project on few random directions. The idea of reducing the dimensionality of a data set using random projections goes back to the Johnson-Lindenstrauss lemma (Johnson & Lindenstrauss, 1984), which says that the pairwise distances between $n$ points are approximately preserved upto $1 \pm \varepsilon$, when they are projected to a random $O(\log n/\varepsilon^2)$ dimensional

subspace. This projection can be achieved essentially by projecting to those many randomly chosen directions. In general, random projections are an important tool to preserve geometry and random directions serve as a good control set of directions. We implement the idea of using median sampling along a few random directions. We call this RandP in experimental results section (Section 6).

## 4.2 Generating balanced dataset using data augmentation

Since predicting Sports73 and Sports100 datasets class distribution is a real challenge, the model finds it hard to retain knowledge learned so far. The problem is exacerbated as the datasets are class imbalanced. The most popular continual learning research focuses on experiments with balanced datasets such as MNIST, CIFAR100, ImageNet, etc. In those datasets, the number of data points is equal between classes, whereas the number of data points in Sports73 and Sports100 is not equal. Therefore, we have proposed to use KeepAugmentation (Gong et al., 2021) algorithm to make each task $t$ of Sports73 and Sports100 balanced during training the feature extractor. Although a substantial amount of work has been devoted to enhancing the variety of exemplars kept in memory through data augmentation, but less work has been devoted to improving the robustness of feature extractor using the technique. Data augmentation improves the classifier's generalization performance to achieve the highest efficiency on learned tasks. First, from a task $t_i = (\mathcal{C}^i, \mathcal{D}^i)$, identify the class $c_j^i \in \mathcal{C}^i$ that has the highest number of images among all the classes in $\mathcal{C}^i$. Let $|c_j^i|$ denote its size. Then, apply the KeepAugmentation algorithm on images of all other classes in the task $t_i$ except for class $c_j^i$, and generate the required number of images $r_k^i$ to make a class $c_k^i$ balanced with respect to the class $c_j^i$, where $k \in [1, m_i], k \neq j$. Therefore, the required number of additional augmented data points to balance the classes can be mathematically expressed as $r_k^i = |c_j^i| - |c_k^i|, k \in [1, m_i], k \neq j$. Let $I_k^i$ denote the $r_k^i$ new images generated for class $c_k^i$ and $\mathcal{D}_k^i$ represent the original training dataset of class $c_k^i$. Then, the augmented dataset for class $c_k^i$ is $I_k^i \cup \mathcal{D}_k^i$. The augmented datasets for each class $c_k^i$ for $k \in [1, m_i], k \neq j$ and $c_j^i$ are then used to train the feature extractor.

---

**Algorithm 1:** Sampling of exemplars using median points

---

1 **Function** PBES_Sampler$(X, m)$:
    **Input:** Image training set $X = \{x_1, \ldots\ldots, x_n\}$ of class $c$, $m$ – number of exemplars to
           output
    **Output:** $P$ exemplars

2    **if** *n is odd and m is even* **then**
3        $p = \lceil m/2 \rceil + 1$
4    **else**
5        $p = \lceil m/2 \rceil$
6    Compute the $p$ principal directions of $X$ using PCA
7    $R \leftarrow X$
8    **for** $i \leftarrow 1$ *to* $p$ **do**
9        $L_i = $ Sort points of $R$ along $i$th principal direction
10        **if** $|R|$ *is even* **then**
11            Let $x_1, x_2$ be the lower and higher median points from $L_i$
12            $P \leftarrow P + \langle x_1, x_2 \rangle$
13            $R \leftarrow R \setminus \{x_1, x_2\}$
14        **else**
15            Let $x_1$ be the median point from $L_i$
16            $P \leftarrow P + \langle x_1 \rangle$
17            $R \leftarrow R \setminus \{x_1\}$
18    **return** first $m$ points of $P$

---

## 5 Experimental setup

Experiments used a NVIDIA RTX 3070 GPU, with 8 GB VRAM, 16 GB RAM and PyTorch 1.8.0.

## 5.1 DATASETS

For our work, we used the Sports73 and Sports100 datasets imported from Kaggle. These datasets have images from the sports domain and have 73 and 100 sports classes respectively. They have classes such as swimming, badminton, etc. Sports images were taken indoors and outdoors in diverse lighting conditions. The datasets are divided into train, test and val datasets, and contain 11,365 and 13,572 training images; 365 and 500 testing images; and 365 and 500 validation images respectively. All images are 224×224 pixel sized. Randomly selected classes with task sizes of 5, 10 and 20 (number of randomly selected classes at each task) were used in the experiments. Both the datasets are imbalanced and a histogram visualizing this imbalance for the Sports73 dataset is displayed in Appendix C. Apart from these two datasets, we have done experiments on the Tiny ImageNet dataset to show that our approach also works on a balanced dataset.

## 5.2 IMPLEMENTATION DETAILS

For feature extraction and representation learning ResNet32 (He et al., 2016) model was used. We start our experiments with a learning rate of 2.0 (Rebuffi et al., 2017), and at each epoch, the learning rate will be decreased by 20% on $49^{th}$ and $63^{rd}$ epochs, and the weight decay will be 0.0001 (Rebuffi et al., 2017). The memory size was 550 for Sports73 and 800 for Sports100. Exemplars account for about 4% of the highest number of images among all the classes (Rebuffi et al., 2017), except for the experiments where we test the effect of varying the number of exemplars on performance (Section 6.3). In order to reduce the training and representative memory requirements the datasets were resized to 96×96 pixels, and a batch size of 32 was used for each experiment.

### EVALUATION METRICS

After each task, we evaluate the model by testing it on all the test classes we observed up to that point. Let $a_i$ denote the accuracy obtained in the test after task $i$. We use the metrics, (1) *Average accuracy*, which is defined as the average of accuracies upto task $t$, i.e., $Average\ Accuracy(A_t) = \frac{1}{t}\sum_{i=1}^{t} a_i$ (If $t = T$, then the average accuracy $A_T$ denotes the average training accuracy over all the tasks.), and, (2) *Last accuracy* which is the final accuracy observed after training all the tasks $T$, i.e., $Last\ Accuracy = a_T$. To determine these numbers, three trials were conducted, and the average of each value defined above was computed over the three trials.

## 6 EXPERIMENTAL RESULTS

In our experiments, we first compare our approach with the well-known state-of-the-art approaches such as Rainbow Memory(Bang et al., 2021), GDUMB(Prabhu et al., 2020), and iCaRL(Rebuffi et al., 2017). Further, we compare our approach to two other approaches, Fine-tune and Upper bound. As part of the Fine-tune approach, only the new class images are used for training, and exemplars are ignored. Cross-entropy loss is applied to the new class images, and distillation loss is not taken into consideration. Thus, the number of exemplars is zero. The fine-tune approach is taken as a lower bound for our problem (in terms of performance). In the lower bound, the exemplars are not stored in memory, and the training for all the tasks happen without considering any exemplars. The Upper bound approach considers all the images seen so far at every task and applies cross-entropy loss. In short, the number of exemplars per class is equal to the number of images for that particular class. In other words, it represents the upper bound for our problem (in terms of performance). We have also compared our results with random projections. By conducting ablation studies, we discuss the effectiveness of our exemplar selection in Section 6.2 of this paper.

## 6.1 COMPARATIVE STUDY WITH OTHER EXISTING TECHNIQUES

The final (Last) accuracy and average (Avg) accuracy for each increment task are summarized in Table 1. We observe that the accuracy of the CL approach changes depending on the task size. For a certain number of training classes to learn, a lower task size will result in more incremental tasks, increasing the likelihood of catastrophic forgetting. In contrast, training many classes for every incremental task is likewise a tough challenge for bigger task sizes. In particular, we discover that Fine-tune suffers from a severe catastrophic forgetting problem, with Last and Avg accuracy being

drastically worse than they are with the Upper bound. This is because there is limited training data for every learned task throughout the incremental learning process.

In Figures 3 and 4, we have displayed the accuracy examined at the end of every task with task sizes of 10, 15, and 20. Our approach achieves better results than state-of-the-art methods in continual learning. The performance difference between our approach and the Upper bound is minimal. The results for the Tiny ImageNet dataset are shown in Appendix B.

Table 1: Average and Last task accuracy of Sports73 and Sports100 datasets with task size 5, 10 and 20

| Datasets | Sports73 | | | | | | Sports100 | | | | | |
|---|---|---|---|---|---|---|---|---|---|---|---|---|
| Task size | 10 | | 15 | | 20 | | 10 | | 15 | | 20 | |
| Accuracy | Avg | Last | Avg | Last | Avg | Last | Avg | Last | Avg | Last | Avg | Last |
| | | | | | | | | | | | | |
| RM | 29.01 | 16.71 | 30.87 | 24.65 | 34.95 | 25.47 | 46.67 | 25.2 | 48.44 | 31.6 | 55.68 | 45 |
| iCaRL | 36.34 | 16.25 | 45.79 | 20.26 | 44.54 | 20 | 33.9 | 19.6 | 30.09 | 11.80 | 44.86 | 30.8 |
| GDUMB | 25.83 | 11.23 | 24.51 | 13.15 | 23.49 | 14.79 | 42.72 | 23.4 | 39.83 | 23.6 | 38.19 | 23.2 |
| RandP (Ours) | 54.49 | 37.5 | 52.43 | 37 | 62.43 | 39.6 | 51.74 | 38.1 | 55.40 | 37.52 | 58.93 | 46.4 |
| PBES (Ours) | **62.51** | **41.5** | **62.68** | **41.87** | **68.27** | **46.75** | **59.44** | **43** | **58.22** | **42.09** | **63.97** | **53** |

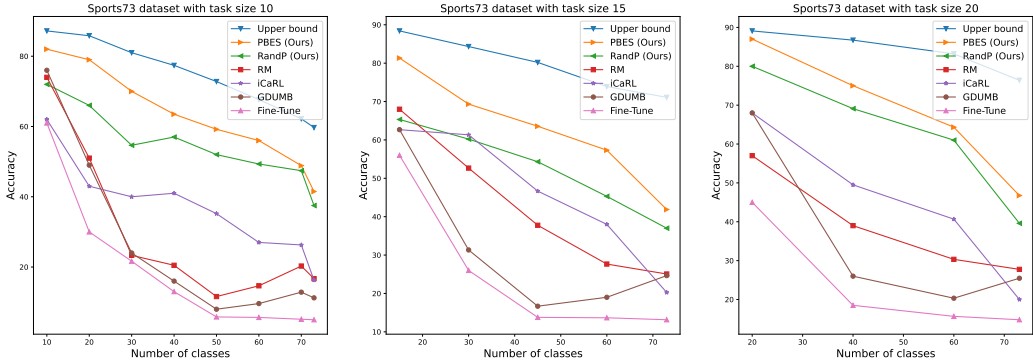

Figure 3: Testing accuracy of Sports73 with task size 10, 15, and 20

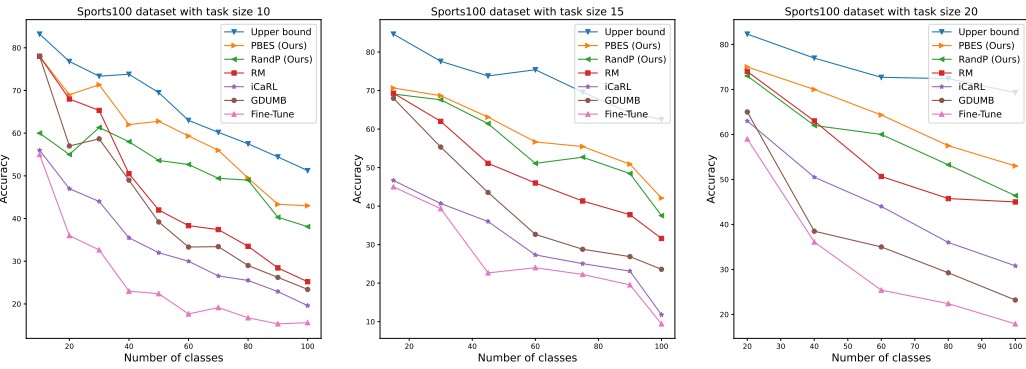

Figure 4: Testing accuracy of Sports100 with task size 10, 15, and 20

## 6.2 ABLATION STUDY

Now, we will study our approach's components and illustrate their influence on the final accuracy. The number of exemplars is kept fixed for these studies. Our approach has two modules 1) module-1: our proposed algorithm PBES, and, 2) module-2: our approach to integrate the KeepAugment algorithm (Gong et al., 2021) for data augmentation. In particular, we compare and contrast the following approaches: (1) **baseline:** select exemplars using the herding algorithm. (2) **PBES**

**sampler without KeepAugmentation :** PBES sampler for exemplar selection but no data augmentation. (3) **baseline and KeepAugmentation method:** Herding algorithm for exemplar selection and KeepAugmentation for data augmentation. (4) **Ours:** PBES sampler for exemplar selection and KeepAugmentation for data augmentation. Each component ( PBES and KeepAugmentation) discussed in this paper, when compared to the baseline, improve performance. We find that when we integrate all the modules of the recommended strategy, our results significantly outperform the baseline. Since there is significant class imbalance present in the Sports73 and Sports100 datasets, where the number of data points in training data varies in the range [98, 191] among sports classes, we also conclude that our training regime gives significant performance improvements for such datasets. The datasets also have huge variance and the proposed strategy performs well even in this case.

Table 2: Average accuracy of different methods on Sports73 and Sports100 datasets with task size 20

| Method | Sports73 | Sports100 |
|---|---|---|
| baseline | 44.86 | 44.54 |
| PBES sampler without KeepAugmentation | 61.2 | 63.56 |
| baseline with KeepAugmentation | 60.67 | 61.28 |
| PBES sampler with KeepAugmentation | 63.97 | 68.27 |

## 6.3 INFLUENCE OF EXEMPLARS SIZE

This section shows the effect of exemplars size on the performance. To test this we vary the total number of stored exemplars $M$ in $\{550, 700, 850, 1000\}$ for Sports73 and $M$ in $\{800, 1000, 1200, 1400\}$ for Sports100. Our average accuracy using the Sports73 dataset with a task size of 20 is displayed in Table 3a and in Table 3b for the Sports100 dataset. For both approaches, improved performance is seen when more exemplars are used. Exemplar memory capacity is amongst the most crucial components for a continual learning systems, especially in an offline environment. We discover that our proposed technique performs better than the baseline for any given number of exemplars. This improvement over baseline becomes more apparent as the number of allowed exemplars decreases, i.e., our method responds much better under a memory crunch.

Table 3: Influence of Exemplars Size for (a) Sports73 dataset & (b) Sports100 dataset

(a)

| $M$=550 | $M$=700 | $M$=850 | $M$=1000 |
|---|---|---|---|
| 68.27 | 71.43 | 75.1 | 78.3 |

(b)

| $M$=800 | $M$=1000 | $M$=1200 | $M$=1400 |
|---|---|---|---|
| 63.97 | 66.4 | 69.2 | 72.23 |

## CONCLUSIONS

This study devised a unique sampling technique that selected representative data from a stream of incoming training data in an effort to reduce the catastrophic forgetting problem of continual learning. Further, a learning regime was presented, employing the data augmentation method, which aids in creating a balanced training set from an unbalanced dataset. Both on complex Sports73 and Sports100 datasets, we discovered that our proposed strategy improves performance over state-of-the-art methods, particularly when the number of new sports classes included at each iteration increased. This points to the feasibility of continual, large-scale learning in practical systems.

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

## A    PROOF OF LEMMA 4.1

*Proof*: When $n$ is even, the algorithm repeatedly selects 2 points from each of the $t$ principal directions as $|T|$ remains even throughout. Thus $2t$ points are selected in all. On the other hand, when $n$ is odd, the algorithm will select 1 point from the first principal direction but 2 from each of the subsequent ones, i.e., for $i = 2, 3, \ldots, t$, since $|T|$ becomes even after the first iteration and remains so in subsequent iterations. So, $1 + 2(t-1) = 2t-1$ points are selected in all. The claim can now be verified by examining all the four possible cases, i.e., $n$ even/odd, $m$ even/odd.    ∎

## B    EXPERIMENTAL RESULTS FOR TINY IMAGENET (BALANCED) DATASET

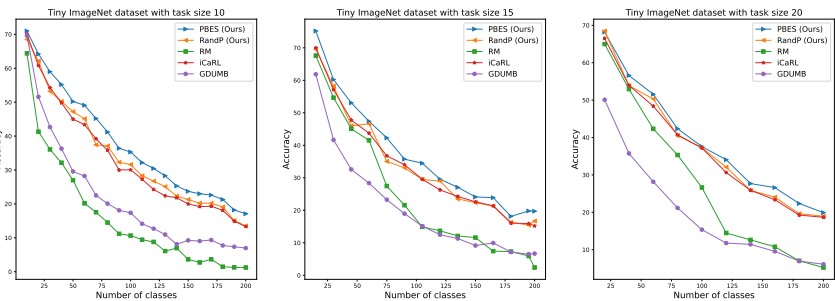

Figure 5: Testing accuracy of Tiny ImageNet with task size 10, 15, and 20, and memory size 2000

## C  HISTOGRAM TO VISUALIZE CLASS IMBALANCE IN SPORTS73 DATASET

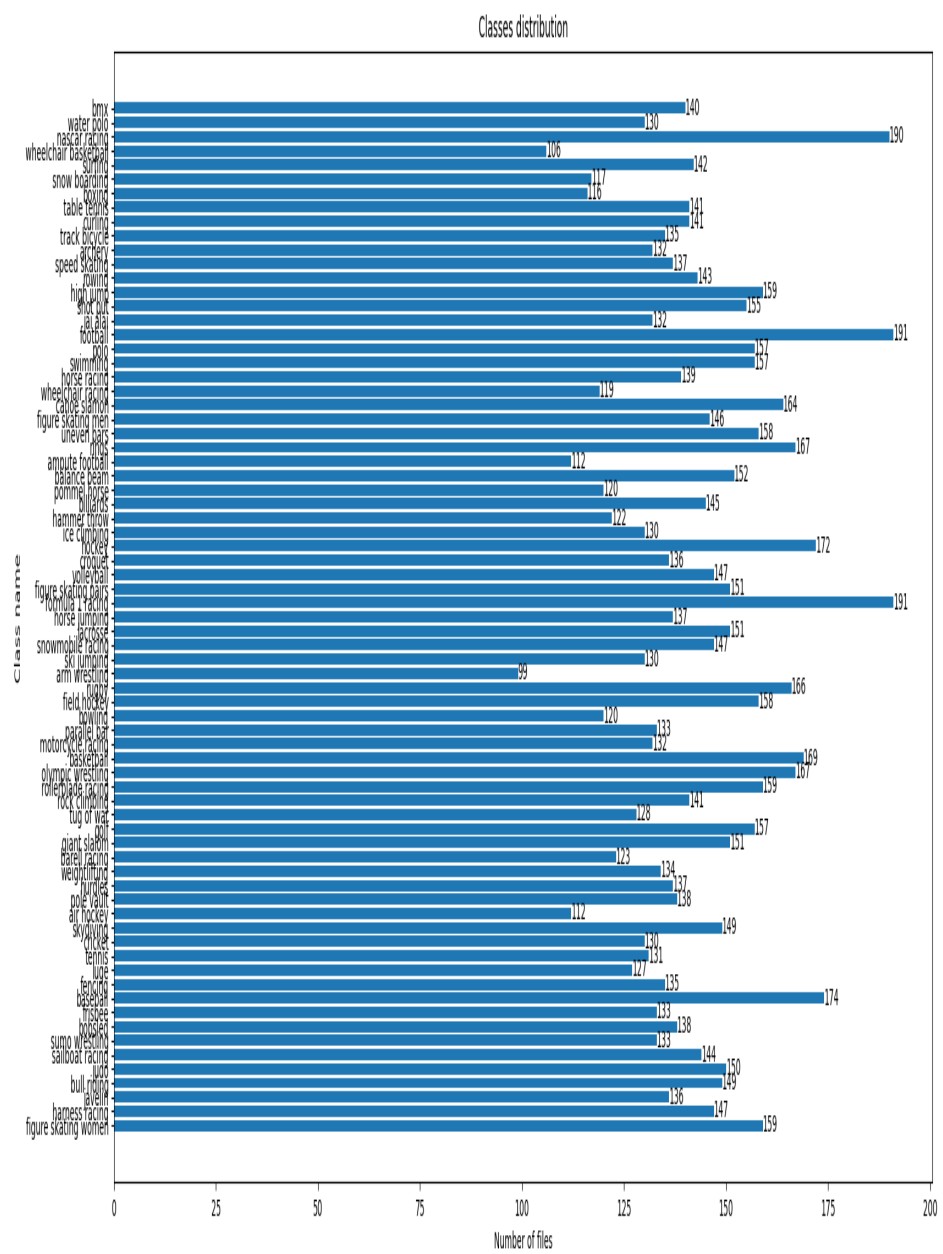

Figure 6: Histogram of Sports73 classes

