# OpenReview forum: "PBES: PCA Based Exemplar Sampling Algorithm for Continual Learning"
_ICLR.cc/2023/Conference — Submitted to ICLR 2023_

### Official Review · Reviewer_JcYz · 2022-10-19

**Confidence:** 4
**Correctness:** 2
**Technical Novelty And Significance:** 2
**Empirical Novelty And Significance:** 1
**Recommendation:** 3

**Clarity, Quality, Novelty And Reproducibility:**

- Clarity : The paper has some ambiguous parts for experimental details, as mentioned in the weakness section.
- Quality : The presentation of the paper is not that neat.
- Novelty : The idea of using PCA to sample exemplars seems novel.
- Reproducibility : While the authors do not provide their source code, the implementation details for the main framework seems sufficient. However, the implementation details for the baseline algorithms should be added, since the authors conducted the experiments for Sports 73 and Sports 100 for the first time.

**Strength And Weaknesses:**

- Strengths
    - The proposed framework demonstrates better performance than other baselines for the newly conducted benchmarks, Sports 73 and Sports 100.
    - The idea of sampling the most unaffected data along the principal axis seems novel and interesting.

- Weaknesses
    - The writing quality of the paper is far behind the bar of top-tier conferences like ICLR.
        - For instance, what is the point of the first paragraph of the introduction section? The paragraph consists of intro to CF, data distribution and class imbalance, the goal of this research, and etc. The paragraph should be split into multiple paragraphs to represent the crucial points for the authors' claim.
        - Formatting issues
            - Missing numbering for some subsections (e.g., Evaluation Metrics) and sections (e.g., Conclusion).
            - Inconsistency of the format for the references.
    - Datasets
        - Is there any quantitative evidence that sports datasets have higher variance than other common datasets, such as CIFAR-100 and ImageNet, which are widely used for continual learning benchmarks? For me, the claim is not that convincing since many sports-based video datasets, such as UCF-101, or Fine-gym, do not have large intra-class variance. While I’m not an expert on the Sports 73 or Sports 100 datasets, I’m not sure why such datasets have high variance and are expected to have many outliers. Actually, when I schemed the Sport 100 datasets on the Kaggle page, it seems the datasets have less variance than CIFAR or ImageNet, since the viewpoint of camera and poses of actors seem pretty similar to each data in the same class. The claim that “Sports images were taken indoors and outdoors in diverse lighting conditions” in the Section 5.1 also seems inappropriate since the augmenting contrast is one of the widely used augmentation strategies, and thus the diverse lighting will be considered when we train datasets like ImageNet with strong data augmentation.
    - Possible discussion in the Methods section
        - Selection of exemplars
            - As far as I understand, the sampled exemplars by $i$th principal direction and $j$th (where $i < j$) direction are not related to each other much. Then, the median points sampled by $j$th direction could be the outliers for $i$th direction, since two different principal axes are orthogonal. While it might be the extreme case, I think it might happen quite often when $m$ increases since the data embeddings used for sampling lie in the high-dimensional space. The claim of the paper could be improved if the authors provide any mathematical proof or intuitive explanation for this.
            - In the Abstract section, there is a sentence "herding, which is a way to select a random looking sample by a deterministic algorithm". However, the proposed PCA-based framework is also almost deterministic given the dataset except for the case that some directions share the same singular values. Moreover, what do the authors mean that herding selects a random looking sample, while it selects the samples closest to the mean class vector?
        - Generating balanced dataset using data augmentation
            - What is the reason for generating data only for $r_k^i = \|c_j^i\| - \|c_k^i\|$? It might be better to generate $\|c_j^i\|$ data for each class using KeepAugment and replace $\|c_k^i\|$, rather than just generating $\|r_k^i\|$ to boost the regularization effect.
    - Experimental results are not sufficient to support the effectiveness of the proposed framework.
        - The combination of other sampling algorithms and the diverse data augmentation strategy must be studied more deeply.
            - According to Table 2, it seems KeepAugment highly affects the performance. Then, it is worth checking the performance of other baselines (RM and GDumb) equipped with KeepAugment. Moreover, the good results on TinyImageNet in Figure 5 in Appendix B also might be due to the effect of KeepAugment, since the difference is marginal. Therefore, the effectiveness of the proposed algorithm for the balanced datasets is not convincing. It might be better to show the results on CIFAR-100 datasets, which is a widely used benchmark for class incremental learning. Training details for other baselines on Sports 73 and Sports 100 datasets are also missing.
            - What if the model employs different data augmentation strategies (e.g., CutMix, etc., ), rather than KeepAugment?
        - It is unclear whether improvement in performance is driven by class-balancing or augmented data.
            - What is the reason for generating data only for $r_k^i = \|c_j^i\| - \|c_k^i\|$? It might be better to generate $\|c_j^i\|$ data for each class using KeepAugment and replace $\|c_k^i\|$, rather than just generating $\|r_k^i\|$ to boost the regularization effect.
    - Minor
        - Need consistency for the term “Principal Component **a**nalysis” and “Principal Component **A**nalysis”
        - It seems the columns of Table 2 should be exchanged with each other, according to Table 1.

**Summary Of The Paper:**

This paper addresses the problem of continual learning with a novel exemplar selecting algorithm.
The authors claim that selecting the exemplars based on PCA-based direction well addresses the catastrophic forgetting problem of the datasets with high intra-class variance.
The experimental results on various benchmarks show the effectiveness of the proposed framework.

**Summary Of The Review:**

Despite the paper studies some novel concepts with respect to PCA-based exemplar sampling, I hardly agree that the current version of the submission meets the bar of ICLR, since there are some missing experiments as stated in the weakness section and the writing quality should be improved.

---

> ### Author Response · Authors · 2022-11-19
> **Response to reviewer JcYz**
>
> We thank you for your comments and feedback. In addition to the general updates, we address your concerns here.
> >**Response to “For me, the claim is not convincing since many sports-based video datasets, such as UCF-101, or Fine-gym, do not have large intra-class variance...”**
> Images in ImageNet and CIFAR100 are of everyday objects, and these objects have very limited pose angles. However, sports datasets are of players playing sports, and therefore the poses are much higher. As an example, a sportsman playing badminton has a wide range of motion of his/her hands and other body parts, unlike non-living things that have limited ranges of motion.
>
> >**Response to “As far as I understand, the sampled exemplars by ith principal direction and jth (where i<j) direction are not related to each other much. Then, the median points sampled by jth direction could be the outliers for ith direction, since two different principal axes are orthogonal...”** We thank the astute reviewer for noticing this fact. Indeed, it is possible that the outliers along the projection on the first principal direction are median points along the second one. However, this is of no consequence to our sampling method. Simply because a point is an outlier along the first principal direction does not mean it should be ignored as it may be significant due to its position along the second principal direction. Arguably, one can think of assigning more weight to the first principle direction than the second one, and these are refinements of our basic scheme that we are working on. In future iterations, we will provide mathematical justification of this basic fact - however, we feel that this fact, though true, does not affect in any way the sampling algorithm or its performance.
>
> >**Response to “In the Abstract section, there is a sentence "herding, which is a way to select a random looking sample by a deterministic algorithm". However, the proposed PCA-based framework is also almost deterministic given the dataset except for the case that some directions share the same singular values...”** The reviewer correctly observes that our method is a deterministic one. We did not mean to say that our method is superior to a randomized method, and neither did we mean that our method selects a “random-looking sample”. The herding algorithm is also deterministic - and we meant to say, based on the experiments of [1], that it seems to perform only as well as random sampling.  We agree that the statement that the herding selects a random-looking sample (given that it selects by distance from the mean) is not entirely intuitive.
>
>
> >**Response to “What is the reason for generating data only for rki=|cji|−|cki|? It might be better to generate |cji| data for each class using KeepAugment and replace |cki|, rather than just generating |rki| to boost the regularization effect.”** The reason for generating data only for rki=|cji|−|cki|is to balance the imbalanced Sports73 and Sports100 datasets. As the first step in a task ti = (Ci, Di), determine which of the classes in Ci has the most images. Let |cij| denote its size. Afterward, apply the KeepAugmentation algorithm to all other classes in the task ti except for class cji, and generate the required number of images rki to make a class cki balanced with respect to class cji.
> By replacing cji data with cki data, it means that the original data, cji, is discarded and replaced with the augmented data. It is not wise to discard the original data and only use the augmented data to train a DNN model.
>
>
> >**Response to “What if the model employs different data augmentation strategies (e.g., CutMix, etc., ), rather than KeepAugment?”** We only tried the KeepAugment[2] algorithm and presented the results. We haven’t tried the other data augmentation approaches.
>
> >**Response to “It is unclear whether improvement in performance is driven by class-balancing or augmented data.”** In our approach the class balancing relies on augmentation, so we couldn’t possibly have a class balancing, but no augmentation. However, our ablation studies, do our performance evaluation without any class-balancing or data augmentation.  We tested the following experiment setups in our ablation studies:
> 1. baseline
> 2. PBES sampler without KeepAugmentation
> 3. baseline with KeepAugmentation
> 4. PBES sampler with KeepAugmentation
> Our results show that even our PCA-based median sampler solely outperforms the existing SOTA algorithms.
>
>
> >**References**
> 1. Javed, K., & Shafait, F. (2018, December). Revisiting distillation and incremental classifier learning. In Asian conference on computer vision (pp. 3-17). Springer, Cham.
> 2. Gong, C., Wang, D., Li, M., Chandra, V., & Liu, Q. (2021). Keepaugment: A simple information-preserving data augmentation approach. In Proceedings of the IEEE/CVF conference on computer vision and pattern recognition (pp. 1055-1064).

---

### Official Review · Reviewer_Gq7Z · 2022-10-22

**Confidence:** 4
**Correctness:** 2
**Technical Novelty And Significance:** 1
**Empirical Novelty And Significance:** 1
**Recommendation:** 3

**Clarity, Quality, Novelty And Reproducibility:**

The Paper is written mostly well, with a bit of unnecessary verbosity in the introduction. The concepts explained are basic  It could have benefited from more visuals and be more compact.


**Strength And Weaknesses:**

The proposed technique includes a simple PCA-based dimension reduction and mapping the data along these dimensions and sample data with enough diversity. With this technique, outliers are not included in the training data. There has been a decent amount of experimentation and details provided.  I have seen some drawbacks in the paper, though.

1. How big of a problem is this? What is the real-life use case and significance of the problem, and the innovation aspect (scope/impact) is not clear?
2. The proposed solution is simple, but I would be surprised if something along these lines has never been tried (I did not do a thorough literature survey, but it seems more like a simple use case of PCA). With this regard, I think the paper lacks novelty.
3. There can be more depth and rigor in the experimentation. For example, PCA provides good strength in eliminating outliers. It sort of smooths out the training data—like a filter. However, it comes up with removing more edge/nuance cases in a data set as well. Usually, these samples are less in population and lost in the PCA analysis—often considered as noise. However, they are very valuable with respect to providing unique information for the training. It is not clear what the trade-off is here in this case and how we can address selecting such under-represented groups in this framework.

**Summary Of The Paper:**

Training machine learning on streaming data sets has been an important problem with the widespread use of online systems. This often requires selecting specific examples to update your model. The paper proposes a PCA-based exemplar sampling algorithm.


**Summary Of The Review:**

I don't think that the paper is strong enough to be at ICLR. From experimentation to justifying the importance of the problem, the novelty in the proposed solution, there are many gaps.

---

> ### Author Response · Authors · 2022-11-18
> **Response to reviewer Gq7Z**
>
> We thank you for your comments and feedback. In addition to the general updates, we address your concerns here.
>
> >**Response to “How big of a problem is this? What is the real-life use case and significance of the problem, and the innovation aspect (scope/impact) is not clear?”** As the sports domain has more variance in the dataset, it was selected for the study. Images were captured from different angles, poses, and lighting conditions. Indeed, almost any outdoor game has lots of degrees of freedom and it is possible to find players doing various activities over the course of the game. This is the cause of the variance.
> In addition, the sports domain has not been explored in a continual learning setting. Therefore, research on the sports domain classification problem is a novel problem.
> Even so, our proposed contributions such as PCA-based sampling and a training regime for balancing the dataset with KeepAugment[1] algorithm, are not tailored to the sports domain and can be applied to other domains as well. Experiments on Sports73 and Sports100 datasets were conducted just to show the efficacy of our contributions.
>
> >**Response to “The proposed solution is simple, but I would be surprised if something along these lines has never been tried (I did not do a thorough literature survey, but it seems more like a simple use case of PCA). With this regard, I think the paper lacks novelty.”** As per our literature review, we only found a few papers [2, 3] that use the PCA algorithm to sample data. Even though the solution may seem simple, it outperforms other state-of-the-art techniques at preventing catastrophic forgetting. Moreover, the literature also contains some sampling algorithms that seem quite simple yet effective, such as GDUMB [4], which selects examples in a greedy manner. In fact, it can be argued that simple and effective techniques might be preferable to more intricate techniques for their ease of implementation.
>
> >**Response to “There can be more depth and rigor in the experimentation. For example, PCA provides good strength in eliminating outliers. It sort of smooths out the training data—like a filter. However, it comes up with removing more edge/nuance cases in a data set as well. Usually, these samples are less in population and lost in the PCA analysis—often considered as noise. However, they are very valuable with respect to providing unique information for the training. It is not clear what the trade-off is here in this case and how we can address selecting such under-represented groups in this framework.”** Given that the budget for exemplars is a small number, virtually any method will be forced to ignore edge/nuanced cases. For example, in the herding method, which selects the closest points to the mean, the far-off points are ignored. In general, this is a good strategy as the method is trying to work well on the bulk of the cases, and is perhaps not so good on the corner cases, but that is a trade-off to be good on performance.
> As for the use of the PCA method, we select median points, looking at the projection in a certain principal direction. In theory, even points that do not lie in the subspace spanned by the first few principal directions could be selected still.
>
>
> **References**
> 1. Gong, C., Wang, D., Li, M., Chandra, V., & Liu, Q. (2021). Keepaugment: A simple information-preserving data augmentation approach. In Proceedings of the IEEE/CVF conference on computer vision and pattern recognition (pp. 1055-1064).
> 2. Lih-Yuan Deng, Ching-Chi Yang, Dale Bowman, Dennis K.J. Lin, and Henry Horng-Shing Lu. Big data model building using dimension reduction and sample selection. Personal communication.
> 3. Wang, H., Yang, M., & Stufken, J. (2019). Information-based optimal subdata selection for big data linear regression. Journal of the American Statistical Association, 114(525), 393-405.
> 4. Prabhu, A., Torr, P. H., & Dokania, P. K. (2020, August). Gdumb: A simple approach that questions our progress in continual learning. In European conference on computer vision (pp. 524-540). Springer, Cham.

---

### Official Review · Reviewer_RPyF · 2022-10-23

**Confidence:** 3
**Correctness:** 3
**Technical Novelty And Significance:** 2
**Empirical Novelty And Significance:** 2
**Recommendation:** 5

**Clarity, Quality, Novelty And Reproducibility:**

In general, the paper has a complete structure, clear thinking and some innovation.

**Strength And Weaknesses:**

Strength：
In general, the article has a complete structure and clear thinking.
The experimental setting is reasonable, experiments verified the proposed algorithms on Sports  datasets.

Weaknesses：
The description of the proposed method is not detailed enough, and the innovation point is not outstanding enough.
This paper mainly focuses on the effect of the strategy, and it is suggested to add more theories to support the strategy, so that the method can be more explainable.
The paper points out that the median is more suitable than the mean. It is suggested to carry out relevant experimental demonstration in the experimental part.


**Summary Of The Paper:**

 In this paper, the other proposes a novel selection approach based on
Principal Component analysis and median sampling. This approach avoids the
pitfalls due to outliers and is both simple to implement and use across various
incremental machine learning models. It also has independent usage as a sampling
algorithm.


**Summary Of The Review:**

Overall, the article has a complete structure and clear thinking. But the content needs to be further improved, including experiment and theory.

---

> ### Author Response · Authors · 2022-11-18
> **Response to reviewer RPyF**
>
> We thank you for your comments and feedback. In addition to the general updates, we address your concerns here.
>
> >**Response to “The description of the proposed method is not detailed enough”**
> The experimental setup used in our study was iCaRL [1]. The Feature Extractor is based on the ResNet18 model, as explained in [1]. To sample exemplars from the data stream in iCaRL, we developed a sampling algorithm based on PCA and median, and this has been provided in detail. Furthermore, we balanced the Sports73 and Sports100 datasets using KeepAugment[2] - a data augmentation technique. So, we believe that we have explained in detail any new ideas that we came up with, and for prior work, we have provided references.
>
> >**Response to “The paper points out that the median is more suitable than the mean. It is suggested to carry out the relevant experimental demonstration in the experimental part.”**
> We have also conducted experiments using PCA with selecting mean points along the principal directions. However, the results were almost identical to those of the other baseline approaches. The reason is that outliers in any data set would have a massive impact if we were to choose mean points along these principal directions. Therefore, even a single outlier in the dataset affects the mean quite dramatically [3], therefore, it is better to sample median data points.
>
>
>
>
> **References**
> 1. Rebuffi, S. A., Kolesnikov, A., Sperl, G., & Lampert, C. H. (2017). icarl: Incremental classifier and representation learning. In Proceedings of the IEEE conference on Computer Vision and Pattern Recognition (pp. 2001-2010).
> 2. Gong, C., Wang, D., Li, M., Chandra, V., & Liu, Q. (2021). Keepaugment: A simple information-preserving data augmentation approach. In Proceedings of the IEEE/CVF conference on computer vision and pattern recognition (pp. 1055-1064).
> 3. Rousseeuw, P. J. (1990). Robust estimation and identifying outliers. Handbook of statistical methods for engineers and scientists, 16, 16-1.

---

### Decision · Program_Chairs · 2023-01-20

**Decision:**

Reject

**Justification For Why Not Higher Score:**

writing/experiments should be improved.

**Justification For Why Not Lower Score:**

n/a

**Metareview: Summary, Strengths And Weaknesses:**

This paper proposes a PCA-based exemplar sampling method in continual/incremental learning. All three knowledgeable reviewers agreed that this paper is not yet ready for publication due to several shortcomings. In particular, many reviewers pointed out that writing and presentation should be improved, and the experimental results are not convincing. The verification using only sports-based video data without using standard benchmark datasets such as CIFAR or ImageNet also raised doubts about the superiority of the proposed method. In addition, some additional concerns on the design choice of the method were also raised by one reviewer, but unfortunately, the authors did not respond, so the proposed method was not sufficiently justified despite some novelty. This AC would like to encourage the authors to resolve all the issues raised this time and submit it to other venue.